# Novel Neonatal Simulator Provides High-Fidelity Ventilation Training Comparable to Real-Life Newborn Ventilation

**DOI:** 10.3390/children8100940

**Published:** 2021-10-19

**Authors:** Joanna Haynes, Peder Bjorland, Øystein Gomo, Anastasia Ushakova, Siren Rettedal, Jeffrey Perlman, Hege Ersdal

**Affiliations:** 1Department of Anaesthesia, Stavanger University Hospital, 4011 Stavanger, Norway; hege.ersdal@safer.net; 2Faculty of Health Sciences, University of Stavanger, 4021 Stavanger, Norway; siren.irene.rettedal@sus.no; 3Department of Paediatrics, Stavanger University Hospital, 4011 Stavanger, Norway; peder.aleksander.bjorland@sus.no; 4Department of Clinical Science, University of Bergen, 5007 Bergen, Norway; 5Research and Development Department, Laerdal Medical, 4002 Stavanger, Norway; Oystein.Gomo@laerdal.com; 6Department of Research, Section of Biostatistics, Stavanger University Hospital, 4011 Stavanger, Norway; anastasia.ushakova@sus.no; 7Department of Pediatrics, Weill Cornell Medicine, New York, NY 10065, USA; jmp2007@med.cornell.edu

**Keywords:** neonatal resuscitation, positive pressure ventilation, respiratory function monitor, deliberate practice, in-situ simulation training, perinatal mortality

## Abstract

Face mask ventilation of apnoeic neonates is an essential skill. However, many non-paediatric healthcare personnel (HCP) in high-resource childbirth facilities receive little hands-on real-life practice. Simulation training aims to bridge this gap by enabling skill acquisition and maintenance. Success may rely on how closely a simulator mimics the clinical conditions faced by HCPs during neonatal resuscitation. Using a novel, low-cost, high-fidelity simulator designed to train newborn ventilation skills, we compared objective measures of ventilation derived from the new manikin and from real newborns, both ventilated by the same group of experienced paediatricians. Simulated and clinical ventilation sequences were paired according to similar duration of ventilation required to achieve success. We found consistencies between manikin and neonatal positive pressure ventilation (PPV) in generated peak inflating pressure (PIP), mask leak and comparable expired tidal volume (eV_T_), but positive end-expiratory pressure (PEEP) was lower in manikin ventilation. Correlations between PIP, eV_T_ and leak followed a consistent pattern for manikin and neonatal PPV, with a negative relationship between eV_T_ and leak being the only significant correlation. Airway obstruction occurred with the same frequency in the manikin and newborns. These findings support the fidelity of the manikin in simulating clinical conditions encountered during real newborn ventilation. Two limitations of the simulator provide focus for further improvements.

## 1. Introduction

The need for neonatal resuscitation is ubiquitous and often unpredictable. Positive pressure ventilation (PPV) of the non-breathing newborn is the cornerstone of resuscitation. In-situ simulation training is widely used to prepare healthcare personnel (HCP) to manage this stressful and time-critical event. Simulation training has shown the potential to change clinical management of babies; however, data to support improved outcomes are limited [1].

PPV is a seemingly simple intervention, which belies the complex interplay of elements necessary for success. Fundamental to ventilation in the non-breathing newborn is the establishment of functional residual capacity (FRC). That can usually be achieved by PPV coupled with positive end-expiratory pressure (PEEP). Mitigating factors that may influence establishing FRC include mask leak and obstruction of the upper airways. Studies of neonatal PPV, using respiratory function monitors (RFMs) to evaluate ventilatory mechanics, highlight the challenges faced by HCPs, with large leaks around the face mask and obstructed upper airways resulting in widely-varying tidal volume (V_T_) delivery [2,3]. Other studies have reviewed the role of RFMs in teaching effective PPV during simulated resuscitation [4,5,6]. Research suggests that HCPs face the same obstacles to effective mask-ventilation as they do in real life [7,8,9,10].

Newer simulators aim for fidelity of approximating clinical conditions. Despite improvements, valid concerns exist regarding the extent to which learning on a simulator can translate into clinical competence [6,11,12]. Specifically, the changing neonatal lung conditions encountered during newborn resuscitation are not replicated by commonly used simulators [13,14]. To be effective, a simulator should closely replicate the clinical situation, in both form and function, and promote management of known hindrances to effective PPV in a way that corresponds to that practised in the clinical environment.

Previous attempts to identify the functional fidelity of neonatal simulators have relied on subjective user feedback rather than measured respiratory parameters [15]. To our knowledge, no existing study has examined the ventilatory mechanics of a neonatal simulator and directly compared them to clinical data from real resuscitations. This study aims to do just that, using the novel high-fidelity manikin NeoNatalie Live™ (Laerdal Medical, Stavanger, Norway). The manikin aims to simulate changing lung compliance encountered during neonatal PPV by using a valve mechanism to alter the physical resistance to lung inflation. Coupling heart rate changes to ventilation performance provides a realistic experience of assessing the effectiveness of PPV. By comparing ventilation parameters and their inter-relationships, along with the occurrence of upper airway obstruction between the manikin and real resuscitations, we aim to demonstrate the functional fidelity of this new simulator.

## 2. Materials and Methods

### 2.1. Study Setting

This prospective, observational study was conducted at Stavanger University Hospital (SUS), Norway. It is the only hospital in the region with both delivery and newborn services, managing approximately 4500 births per annum and providing care for newborns ≥ 23 weeks’ gestational age (GA). Rate of PPV provision at birth is 3.7%, and most neonates are resuscitated by a paediatrician [16]. In some unforeseen resuscitations, PPV is initiated by midwifery or anaesthetic staff. All HCPs receive neonatal resuscitation training according to national resuscitation guidelines. Most PPV is provided using a flow-driven T-piece resuscitator (NeoPuff^TM^, Fischer and Paykel, Auckland, New Zealand).

### 2.2. The Neonatal Simulator

NeoNatalie Live is a newborn simulator, produced with the specific aim of training competence in PPV. Simulated, changing lung compliance and variable heart rate linked to ventilation performance allow HCPs to practise management of newborns with differing degrees of birth asphyxia. Real resuscitation data derived from 1237 newborns informs the algorithm guiding the realistic heart rate response according to PPV provided [17]. An active electrocardiogram allows monitoring of heart rate using the dry-electrode technology NeoBeat^TM^ (Laerdal Medical), replicating the clinical situation. A sensor measures air pressure in the upper airway. Head-tilt detection identifies upper airway closure due to poor positioning. A cry-sound indicates spontaneous respiration and successful resuscitation. Communication with a training application (NeoNatalie Live, Laerdal Medical) allows HCPs to review their performance and gives targeted feedback in any of four scenarios (1–4) of increasing difficulty. Bluetooth^®^ technology allows for the collection of training data in a web log. Figure 1 shows a participant ventilating NeoNatalie Live.

The single NeoNatalie Live manikin used in this study was identified as leak-free internally. Details of this process are available from the corresponding author. Any leak measured by the RFM during mask ventilation was thus attributed to leak at the face mask.

### 2.3. The Respiratory Function Monitor

Each resuscitation bay was equipped with a Newborn Resuscitation Monitor (Laerdal Medical) that recorded airway pressures and gas flow. Sensors were placed between the T-piece and the face mask. Air pressure was measured using a piezoresistive sensor (MPXV5010, Freescale Semiconductor Inc., Austin, TX, USA). The flow sensor (MIM Gmbh, Krugzell, Germany) has negligible resistance and dead space (1 mL), and measures airflow using hot wire anemometer technology. V_T_ is calculated as flow integrated over time. The flow sensor measures both the inflated and expired gas. Expired volume is used as an estimate for V_T_ since mask leak is reported to primarily occur during inflation [9]. Face mask leak is calculated as a percentage of inspired V_T_ from the formula ((V_Tinspired_ − V_Texpired_)/V_Tinspired_) × 100 [9]. The resuscitation monitor has been further described previously [18].

### 2.4. Study Design

This study is part of a comprehensive research set-up at SUS, called Safer Births Bundle [19]. Resuscitation and ventilation data are continuously recorded for newborns of consenting parents, and who are GA ≥ 37 weeks without innate cardiorespiratory anomalies.

Eighteen paediatricians were recruited to this study. Following an individual teaching session with NeoNatalie Live, the paediatricians performed two simulated resuscitation scenarios. The first and easiest scenario (S1—apnoea, normal lung compliance, compensated heart rate) and the most difficult scenario (S4—apnoea, low initial lung compliance and decompensated heart rate), required 30 and 90 s, respectively, of optimal PPV to achieve baby-cry (suboptimal PPV resulted in longer scenario times). The RFM recorded ventilatory parameters during simulated PPV.

Each of these 36 simulated ventilation episodes was paired with a real ventilation episode of similar duration of PPV (±15%), allocated consecutively from the clinical data-pool. This manikin-baby ventilation pairing was made according to the premise that the duration of PPV required to initiate adequate spontaneous respiration is a proxy for the clinical condition at birth. Thus a further 36 clinical ventilation episodes also recorded by the RFM were included. Nineteen neonates received continuous positive airway pressure (CPAP) immediately following cessation of PPV for up to two minutes. No neonate required transfer to the Neonatal Intensive Care Unit for continued CPAP, and none were intubated.

All ventilation was performed with the NeoPuff T-piece resuscitator with standard settings of 8 L/min gas flow and initial 30 cmH_2_O PIP and PEEP of 5 cmH_2_O. PIP could be increased to 35cmH_2_O at the discretion of the HCP. A standard silicone facemask size 0/1 (Laerdal Medical) or the newer snap-design silicone mask size 1 (Laerdal Medical) was used on both babies and manikin.

The 72 ventilation episodes were allocated to one of four groups of 18 PPV sequences (=total PPV time, excluding any pauses > 5 s), according to the recipient of PPV (manikin-M, or baby-B) and the duration of ventilation (≈30 s = short-S, or ≈90 s = long-L).

### 2.5. Data Collection

For each PPV sequence, per individual inflation values of PIP, PEEP, expired tidal volume (eV_T_) expressed as ml/kg and mask leak % were recorded. For manikin V_T_ data, the median birth weight of the 1237 newborns contributing data to the simulation algorithm was used [18]. Additionally, we assessed whether upper airway obstruction occurred. This was defined as minimal inspiratory/expiratory gas flow and V_T_ for three or more consecutive ventilations, despite achieving target PIP, identified from the graphical output of the RFM. The inflations immediately preceding, and/or following, obstruction achieved flows and volumes typical of the whole sequence. This is likely to represent failure of positioning to maintain an open airway [3,20]. Figure 2 shows PIP, eV_T_ and flow curves for one of the clinical ventilation episodes, demonstrating airway obstruction occurring between 25 and 30 s.

### 2.6. Data Analysis

Data analysis was undertaken using SPSS (IBM SPSS Statistics for Windows, Version 26.0. Armonk, NY, USA: IBM Corp) and R project for statistical computing (https://www.r-project.org, accessed on 1 September 2021) version 4.0.4. R package plm version 2.4-1 was used to estimate linear panel models and package dynCorr version 1.1.0. was used to evaluate dynamical correlation. Scatterplots were produced using R package ggplot2 version 3.3.5.

First, continuous data for the ventilatory parameters PIP, PEEP, eV_T_ and leak were summarised using median and interquartile range (IQR) for each of the four groups (manikin short-MS, baby short-BS, manikin long-ML and baby long-BL) and presented using boxplots.

To compare the dynamics of ventilatory parameters between the groups, we used panel data regression analysis with one-way random effects models with a temporal error component. Comparisons were done separately for short and long ventilation sequences. The use of each model was justified by unit root test for stationarity [21]. A Newey and West variance estimator was used to correct for the serial correlation and heteroscedasticity in the residuals [22]. The *p*-values of these comparisons are presented together with the corresponding box plots. For each ventilatory parameter, we present their smoothed trajectories obtained using the LOESS method with a smoothing span of 0.5. Individual ventilations with data from at least five of the 18 ventilated subjects in each group were used for these dynamic trend plots.

Dynamics of the ventilatory parameters between manikin and baby groups were then formally compared using the method of dynamical correlation for multivariate longitudinal data [23]. Pearson correlation analysis is unsuited to this repeated-measures data because the data has a high degree of autocorrelation [24], but in order to place our findings in the context of other research, Pearson correlation coefficients are given, along with scatter plots depicting these correlations. P-values and 95% confidence intervals (CI) for dynamical correlation were calculated using the bootstrap method with 1000 samples.

Chi-Square testing was used to compare the occurrence of airway obstruction between the combined manikin and combined baby groups. A *p*-value < 0.05 was considered statistically significant.

## 3. Results

### 3.1. Ventilations Analysed

Each group (MS, BS, ML and BL) consisted of 18 PPV sequences of a varying number of ventilations. A total of 3256 ventilations were analysed, and distributed as follows: MS 443, BS 475, ML 1160, BL 1178.

### 3.2. Ventilatory Parameters

Median (IQR) values for PIP, PEEP, eV_T_ and leak in the four groups are presented in Table 1.

Figure 3 shows these data as box plots along with the corresponding trend lines estimated as smoothed mean (standard error) for the parameters PIP, PEEP, eV_T_ and leak. PIP was higher in ML compared to BL and greater variation around the central tendencies was seen in the baby-compared to the manikin-groups (Figure 3a,e). Overall, PEEP was lower in manikin ventilation but with a similar variation compared to baby ventilation (Figure 3b,f). eV_T_ was generally lower for the manikin, but values converged towards those observed in the babies at the end of short and long sequences. As with PIP, eV_T_ variation is lower for the manikin than for babies (Figure 3c,g). Median eV_T_ was greater in long sequences compared to short sequences for both manikin and babies. A large variation in leak was observed in all ventilation sequences; variation between short and long sequences for manikin and babies reversed the pattern seen for eV_T,_ with lower median leak in longer sequences (Figure 3d,h).

### 3.3. Dynamical Correlation between Ventilatory Parameters

For all four groups, dynamical correlation between PIP and eV_T_, PIP and leak, and leak and eV_T_ are shown in Table 2. Pearson’s correlation coefficient, r, is given under the corresponding dynamical correlation coefficient, ρ.

To give a visual impression of these correlations, scatter plots are shown in Figure 4. Within the manikin and baby groups, each correlation was either positive or negative. Therefore, to simplify the figure, data for combined groups are shown, i.e., MS + ML and BS + BL.

### 3.4. Obstruction

There was no difference in the occurrence of upper airway obstruction between manikin and babies, with four and seven ventilation sequences, respectively, in which obstruction occurred, *p* = 0.33.

## 4. Discussion

This study compared ventilation of a high-fidelity term neonatal simulator with ventilation of term newborns, matched by a proxy for clinical condition. We aimed to assess the degree to which NeoNatalie Live provides a realistic representation of the experience of ventilating a non-breathing newborn. Our findings of comparable values of four ventilatory parameters, similar inter-relationships between these parameters, and the occurrence of upper airway obstruction to the same degree in manikin and newborn ventilation support the fidelity of the simulated experience of neonatal PPV.

### 4.1. Ventilatory Parameters

Similar median PIP, corresponding to the set value, was generated in all four groups, consistent with previous clinical [10,20] and simulation studies [25,26]. The higher smoothed-mean in ML compared to BL, reflected in the significant p value of the boxplots, was expected, as PIP was sometimes intentionally increased from 30 to 35 cmH_2_O in this group to overcome low lung compliance and achieve visual chest rise. The greater variability of delivered PIP in the baby groups, seen in Figure 3a,e, may result from variation in the clinical condition not being replicated in the simulated setting- for example spontaneous movement of the baby, neonatal respiratory efforts [27,28], or continued stimulation.

Wide variation in PEEP was seen in all groups, and delivered PEEP was lower in the manikin groups. This is consistent with previous clinical [10,20] and simulation data [25].

Using eV_T_ corrected for birth weight for manikin data is unusual and, to our knowledge, has not been described previously. This approach was essential in this study in order to compare simulated and clinical data. The actual weight of the manikin used is 1.54 kg, however, the manikin’s size (length and head circumference) simulates a newborn of around 3 kg birth weight. Therefore, we chose to use the median weight (3.14 kg) of newborns in the study supplying heartrate data [17]. Manikin studies quoting actual eV_T_ [5,11] are difficult to compare to clinical studies quoting eV_T_/kg. We found comparable manikin and neonatal eV_T_s/kg, below and at the lower end of recommended ranges [29] and in line with other reports of neonatal PPV with NeoPuff at standard settings [3,10]. This is a novel and important finding, particularly in light of concerns regarding the unphysiological compliance curves of typical neonatal manikins [14].

We found that higher median volumes were achieved in both manikin and babies when longer ventilation is required. A recent study described a progressive increase in eV_T_ over the first 20 ventilations in term neonates requiring PPV at birth [30]. The authors relate this to the establishment of FRC. Our clinical data may support this. Interestingly, a sharp increase in mean eV_T_ is seen in the dynamic MS plot (Figure 3g, short sequence) and is due to the initially flat and empty manikin lung being filled with air during the first few ventilations before reaching the “air in = air out” stage.

Mask-leak was similar in the short ventilation groups. However, our study confirms previously published data showing both large and highly variable mask leaks during both manikin and neonatal PPV [2,7]. Even experienced HCPs are reported to have large, and often unappreciated, leaks during PPV [12,31,32]. There is, however, a trend towards lower leak in both manikin and baby groups in long sequences versus short. This might imply more successful leak-reducing manipulations given more time to make adjustments.

### 4.2. Correlations between Ventilatory Parameters

Significant dynamical correlations were found between leak and eV_T_ for MS, ML and BL groups. For the other relationships (i.e., PIP and eVT, PIP and leak), no clear correlation was found. This is in contrast to published data where linear or the Pearson correlation are typically used, and thus comparisons with our dynamical data, which compare slopes of the trend lines, are difficult to interpret. A weak, but unquantified, relationship between PIP and eV_T_ has been reported in preterm neonates [2] and a term manikin [31]. A simulation study using a different manikin and a lower set PIP found a strong correlation between PIP and eV_T_ and a moderate negative correlation between PIP and leak [11]. The Pearson’s r for our scatter plots in Figure 4 shows a weak to moderate correlation for PIP/eV_T_ and moderate correlation for PIP/leak. The scatter plots highlight the similar relationships in both manikin and neonatal ventilation, again with a distinct greater variability in the clinical compared to simulation data.

The lack of correlation between PIP and either eV_T_ or leak using a more robust, non-parametric method is perhaps predictable. eV_T_ varies widely in studies using set PIPs [2,10,32]. Similarly, it has been demonstrated that at high gas flows, a set PIP is consistently achieved, independent of mask-leak, unless the latter is very large [13,31]. We did find a strong, significant correlation between eV_T_ and leak:- eV_T_ increases as leak decreases. This is inevitable, given that leak is calculated as the fraction of the difference between inspired and expired V_T_ and inspired V_T_. However, we believe that this does not detract from the probability of a real effect. The consistency of this relationship has been demonstrated in other studies [3,13,30,33].

### 4.3. Obstruction

Defining upper airway obstruction as occurring in ventilations with minimal flow/V_T_ despite adequate PIP likely represents inadequate head positioning to open the airway [2,3]. Our finding of no difference in the occurrence of obstruction in the manikin or babies when experienced paediatricians provide PPV suggests that airway patency is being maintained in similar ways in both groups, with potential for skills learned on the manikin to translate to the clinical scenario.

### 4.4. Limitations in the Fidelity of Simulated Neonatal Ventilation

Despite the similarities, this study highlights two main limitations of NeoNatalie Live. First, the transition between “non-breathing” and “breathing adequately” is very abrupt for the manikin compared to a more gradual change in the babies. This is represented visually in the MS and BS dynamical smoothed-mean PIP and PEEP plots (Figure 3e,f), where both pressures are maintained to the last ventilation for the manikin, whereas for the babies, these values fall. We believe this is due to ventilations with a less tightly applied face mask when evaluating the adequacy of spontaneous efforts in the babies. This pressure fall is mirrored by a simultaneous, considerable increase in leak (Figure 3h).

Secondly, the most difficult scenario 4 in NeoNatalie Live likely combines a low lung compliance derived from severely asphyxiated neonates, with a too rapid increase in heartrate and too short ventilation than that which would be needed to achieve adequate spontaneous ventilation in real life. Additionally, the low manikin compliance is achieved by a closed valve, resulting initially in little or no V_T_ along with no visible chest rise. The valve opens relatively abruptly once sufficient adequate ventilations have been given. The very rapid rise in manikin eV_T_ seen on the trend plot, Figure 3g, for long sequences is clearly different to the neonates in our study, despite the ventilation sequences being paired by duration, and thus, indirectly, by heartrate evolution. Although manikin scenario 4 permits crucial skill training for low-compliant lungs [34,35], there is a disconnect (in heartrate response, ventilation duration and abrupt change in chest rise) with the typical clinical scenario in which these conditions of low lung compliance would likely be encountered.

### 4.5. Strengths and Limitations of the Study

The strengths of this study include the unique design, addressing an issue commonly cited as a limitation to the interpretation of manikin studies [5,6,8,13,33]. The use of experienced paediatricians to ventilate the manikin, and who are also responsible for most real-life PPV, reduces variation which might affect differences between simulated and clinical PPV.

Weaknesses of our study design include the single site setting, limiting generalisability to other institutions. In particular, our use of a flow-driven T-piece resuscitator, rather than a self-expanding bag most commonly employed on a global basis, limits generalisability to other settings.

### 4.6. Future Studies

Future research should focus on addressing the limitations of our study, involving other healthcare settings and looking at simulator fidelity when using a self-inflating bag for PPV. Follow-up studies investigating the training effect of NeoNatalie Live for personnel in different professions, and focusing on training load, have been undertaken and will be reported.

## 5. Conclusions

We compared T-piece PPV of term neonates and a novel, term manikin, paired by a proxy for clinical condition. Our findings of the generation of comparable ventilatory parameters PIP, PEEP, eV_T_ and leak, a consistent inter-relationship between these parameters, and a similar occurrence of upper airway obstruction support the functional fidelity of the simulator. We believe this allows confidence in the ability of NeoNatalie training to foster and maintain PPV skills that can translate into competence in the clinical setting.

## Figures and Tables

**Figure 1 children-08-00940-f001:**
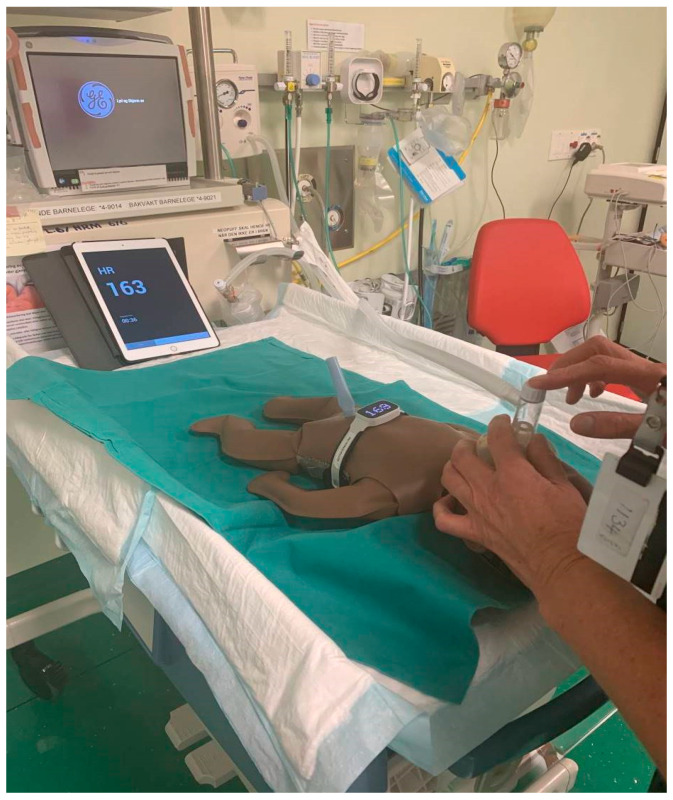
Study participant ventilates NeoNatalie Live with a NeoPuff T-piece resuscitator. The heart rate is clearly visible on both the NeoBeat sensor applied to the manikin and the tablet-device with the training application open.

**Figure 2 children-08-00940-f002:**
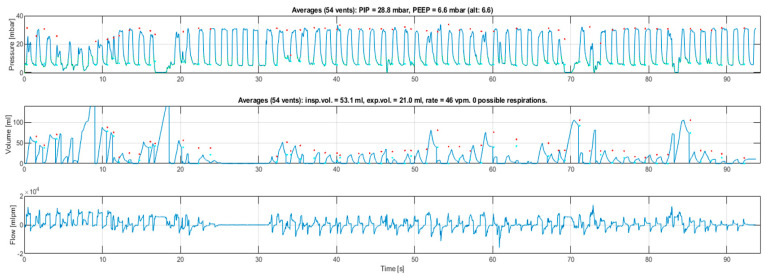
Graphical respiratory function monitor output from a clinical ventilation sequence. The upper curve shows peak inflating pressure, generally maintained around 30 mbar (1 mbar = 1.02 cmH_2_O and the units are used interchangeably in this article). The second curve shows tidal volume (mL); the discrepancy between inflated and expired volumes is due to mask leak. The third curve shows gas flow (mL/min), with positive values indicating flow towards the neonate and negative values indicating flow away. The volume and flow curves disappear while pressure is maintained between 25–30 s, indicating obstruction to gas flow which is rapidly corrected.

**Figure 3 children-08-00940-f003:**
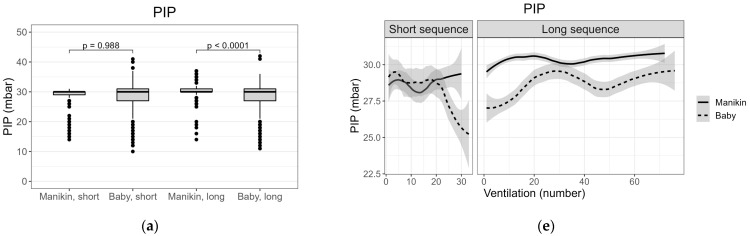
(**a**–**d**) Box plots of median (interquartile range; range) PIP (peak inflating pressure), PEEP (positive end-expiratory pressure, eV_T_ (expiratory tidal volume) and leak for groups Manikin Short (MS), Baby Short (BS), Manikin Long (ML) and Baby Long (BL); (**e**–**h**) Dynamical smoothed mean (standard error) plots of PIP, PEEP, eV_T_ and leak for groups MS, BS, ML and BL.

**Figure 4 children-08-00940-f004:**
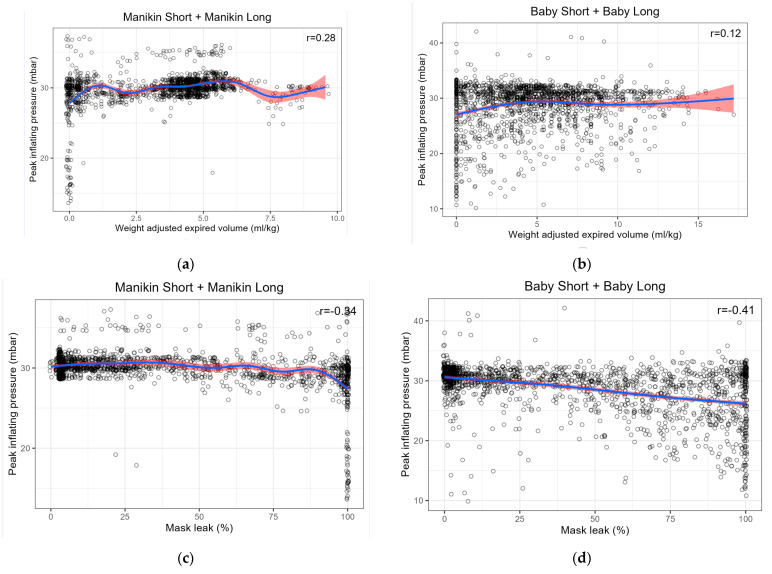
Scatter plots of correlation between (**a**) Peak inflating pressure (PIP) and expired tidal volume (eV_T)_ for combined Manikin Short + Manikin Long (MS + ML); (**b**) PIP and eV_T_ for combined Baby Short + Baby Long (BS + BL); (**c**) PIP and leak for combined MS + ML; (**d**) PIP and leak for combined BS + BL; (**e**) leak and eV_T_ for combined MS + ML; (**f**) leak and eV_T_ for combined BS + BL. Note that Pearson’s r given in the figures differs from those quoted in Table 2 as a result of the combination of manikin and baby groups.

**Table 1 children-08-00940-t001:** Median (interquartile range) ventilatory parameters for the four groups.

Group	Median (IQR)			
	PIP (mbar)	PEEP (mbar)	eV_T_ (mL/kg)	Leak (%)
Manikin Short	30 (1)	3.9 (2.3)	3.5 (3.2)	33.5 (81)
Baby Short	30 (4)	4.9 (2.9)	3.3 (4.6)	50 (80)
Manikin Long	30 (1)	3.7 (2.3)	4.1 (3.2)	20 (57)
Baby Long	30 (4)	4.8 (2.1)	5.0 (4.9)	36 (73)

PIP = peak inflating pressure, PEEP = positive end-expiratory pressure, eV_T_ = expiratory tidal volume.

**Table 2 children-08-00940-t002:** Dynamical correlation and Pearson correlation coefficients for pairwise relationships of PIP-eV_T_-leak for the four groups.

Correlation	Coefficients	Manikin Short	Baby Short	Manikin Long	Baby Long
PIP- eV_T_	Dynamic ρ (95% CI); *p*-value	−0.22 (−0.55;0.34); 0.530	0.27 (−0.31;0.39); 0.866	−0.27 (−0.42;0.16); 0.334	0.14 (−0.16;0.66); 0.262
	Pearson’s r	0.47	0.14	0.14	0.12
PIP-Leak	Dynamic ρ (95% CI); *p*-value	−0.23 (−0.61;0.12); 0.176	−0.22 (−0.47;0.27); 0.702	−0.15 (−0.56;0.18); 0.296	−0.42 (−0.74;0.02); 0.054
	Pearson’s r	−0.51	−0.23	−0.19	−0.49
Leak-eV_T_	Dynamic ρ (95% CI); *p*-value	−0.51 (−0.80;−0.09); 0.020 *	−0.65 (−0.61;0.09); 0.150	−0.47 (−0.61;−0.07); 0.016 *	−0.41 (−0.7;−0.02); 0.032 *
	Pearson’s r	−0.84	−0.58	−0.74	−0.48

* = significantly different from zero at the 0.05 level.

## Data Availability

The data presented in this study are available on request from the corresponding author. The data are not publicly available due to privacy statements made in informed consent obtained from both participating healthcare personnel and parents of studied neonates.

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
