# Peer review of "Novel Neonatal Simulator Provides High-Fidelity Ventilation Training Comparable to Real-Life Newborn Ventilation"

_children, 2021, doi:10.3390/children8100940_

Round 1
Reviewer 1 Report
Overview:
A well- written article on a novel neonatal simulator that provides high-fidelity ventilation training comparable to real-life newborn ventilation. The findings are helpful for the neonatal providers and may increase the use and reliance on manikin/ simulation-based resuscitation for high frequency training to prepare health care personnels.
Major comments:
Why did the authors choose to use the term “non-breathing” newborn? Is this a common term?
Minor comments:
Line 48: “large mask leaks”- consider rephrasing as large leaks around the mask, or in a different manner.
Line 65: How does this manikin aim simulate changing lung compliance? Please explain better to the reader. In this sentence is it ……………“and” couples heartrate changes to ventilation parameters” or is it “by coupling”?
Lines 83 and 86: “heartrate” should be “heart rate”
Line 100: Is there a reference for this?
110: “inflated” should be replaced by either inhaled or inspired.
Line 144: “per inflation values…” is not clear.
Line 147: “assessment was made regarding the occurrence of upper airway obstruction” can be rephrased.
Table 1: Were the data statistically different on median ventilatory parameters between the 4 groups? If so, please add a symbol or mention the p values.
Table 2: Is the asterisk representing significance when comparing all the 4 groups?
Line 319: < …> may be replaced by “….” For non-breathing and breathing adequately.
Reviewer 2 Report
Haynes and colleagues report on the performance of a neonatal simulator that is designed to respond with high fidelity to assisted ventilation. The paper is generally well-written and it contains detailed data and appropriate analyses. Appropriate consent was obtained from parents of human neonates, and also from participating pediatricians.
The Abstract summarizes accurately the contents of the manuscript. However, readers’ attention would be more appropriately directed if the first word, “Ventilation”, were replaced by the term "Face mask ventilation".
Still on line 16, it can be argued that assisted ventilation is not conducted "infrequently", since it is provided to 4 to 10% of newborn infants (per NRP), which is consistent with the rate at the authors’ own institution. Most other procedures associated with neonatal stabilization (intubation, chest compressions, umbilical lines, chest tubes) are performed less frequently, and yet they are not being prioritized for development of a high fidelity manikin. Furthermore, in the introduction it is stated that simulation training is widely used, although it is likely that in most centers it is done less frequently than actual provision of PPV. Perhaps a more accurate argument is that the procedure is critical to neonatal care, and performed frequently but often suboptimally.
In the presentation of Results, to enhance readability, please ensure that in the final formatting of the manuscript each table is fully contained within a single page; the same applies to each figure and its corresponding legend.
Although the data provided strongly support the internal validity of the study, the generalizability may be limited since the algorithm is derived from data in newborns with only mild cardiorespiratory depression (none required additional respiratory support or admission to the NICU); this could be added or clarified in section 4.4.
Also, the providers of ventilation were all pediatricians from a single institution, which presumably narrows the range of ventilation skills relative to those present in the real world – although this is a study strength regarding internal validity, as noted in section 4.5, it is a weakness regarding generalizability, which should be noted in the 2nd paragraph of the same section.
Minor issues:
Should the keywords include numbers? (Lines 32 and 33)
Line 204, correct spelling is “tendencies".
Line 306, correct spelling is “inevitable".
